# First Report of Lysozyme Amyloidosis with p.F21L/T88N Amino Acid Substitutions in a Russian Family

**DOI:** 10.3390/ijms241914453

**Published:** 2023-09-22

**Authors:** Mariya Yu. Suvorina, Elena A. Stepanova, Vilen V. Rameev, Lidiya V. Kozlovskaya, Anatoly S. Glukhov, Anastasiya A. Kuznitsyna, Alexey K. Surin, Oxana V. Galzitskaya

**Affiliations:** 1Institute of Protein Research, Russian Academy of Sciences, 142290 Pushchino, Russia; marrruko@yandex.ru (M.Y.S.); gluktol@gmail.com (A.S.G.); letova_1995@mail.ru (A.A.K.); alan@vega.protres.ru (A.K.S.); 2Federal State Budgetary Educational Institution of Further Professional Education “Russian Medical Academy of Continuous Professional Education” of the Ministry of Healthcare of the Russian Federation, 125993 Moscow, Russia; elena_stepanova99@mail.ru; 3State Budgetary Healthcare Institution “City Clinical Hospital named after V.M. Buyanov of Moscow Healthcare Department”, 115516 Moscow, Russia; 4Tareev’s Clinic of Internal, Occupational Diseases and Rheumatology, Sechenov’s First Moscow State Medical University, 119021 Moscow, Russia; vvrameev@mail.ru (V.V.R.); lidia.v.lysenko@gmail.com (L.V.K.); 5Branch of the Shemyakin–Ovchinnikov Institute of Bioorganic Chemistry, Russian Academy of Sciences, 142290 Pushchino, Russia; 6State Research Center for Applied Microbiology and Biotechnology, 142279 Obolensk, Russia; 7Institute of Theoretical and Experimental Biophysics, Russian Academy of Sciences, 142290 Pushchino, Russia

**Keywords:** genetic testing, hereditary systemic amyloidosis, histology, lysozyme amyloidosis, mass spectrometry

## Abstract

Lysozyme amyloidosis is caused by an amino acid substitution in the sequence of this protein. In our study, we described a clinical case of lysozyme amyloidosis in a Russian family. In our work, we described in detail the histological changes in tissues that appeared as a result of massive deposition of amyloid aggregates that affected almost all organ systems, with the exception of the central nervous system. We determined the type of amyloidosis and mutations using mass spectrometry. Using mass spectrometry, the protein composition of tissue samples of patient 1 (autopsy material) and patient 2 (biopsy material) with histologically confirmed amyloid deposits were analyzed. Amino acid substitutions p.F21L/T88N in the lysozyme sequence were identified in both sets of samples and confirmed by sequencing of the lysozyme gene of members of this family. We have shown the inheritance of these mutations in the lysozyme gene in members of the described family. For the first time, we discovered a mutation in the first exon p.F21L of the lysozyme gene, which, together with p.T88N amino acid substitution, led to amyloidosis in members of the studied family.

## 1. Introduction

Amyloidosis caused by the deposition of lysozyme protein in the form of amyloid (ALys type amyloidosis) is an inherited autosomal dominant disease with incomplete penetrance. The role of lysozyme in the development of amyloidosis was first described in 1932 [1]. At the moment, about thirty cases of this disease are known [2]. The disease is usually manifested by sicca syndrome and lesions of the gastrointestinal tract (GIT), followed by involvement in the pathological process of the kidneys, spleen, peripheral nervous system, and heart. Any organs and tissues, with the exception of the central nervous system, can be involved in the pathological process. The onset age varies from 14 to 50 years [3].

Lysozyme amyloidosis is an autosomal dominant disorder with heterozygous mutations. That is, only one allele of a mutated lysozyme gene is necessary for disease. The first mutations leading to ALys amyloidosis were identified in exon 2 of the LYZ gene (p.Ile74Thr and p.Asp85His) [4]. Then, in 2004, a new missense mutation (p.Trp130Arg) was identified in exon 4. In 2017, another mutation was detected in exon 3 [5]. At the moment, nine different mutations in the LYZ gene that lead to the development of this disease are known. Currently, two variants have been identified that are considered to be non-amyloidogenic for this protein and occur in the population with a frequency of more than 1%: c.263C → A nucleotide substitution in exon 2 (p.Thr88Asn) and c.388T → C nucleotide substitution in exon 4 (p.Trp130Arg) [6]. Since these substitutions do not lead to pathological protein aggregation, they can apparently be classified as an allelic polymorphism with the only exception: if they are not both present in the same gene sequence [7]. Lysozyme mutations and the corresponding amino acid substitutions are presented in Table 1. The locations of the corresponding amino acid residues in the protein structure are shown in Figure 1.

It should be mentioned that there are several zones in the geography of the disease. For example, p.Trp82Arg mutation in exon 2 was first diagnosed in a patient of Italian origin (Piedmont, Italy) and suggests a founder effect for this mutation [8]. Recently, the new mutation p.Asp105Gly leading to ALys amyloidosis was discovered in the LYZ gene in exon 3 [9].

**Table 1 ijms-24-14453-t001:** Mutations in the lysozyme gene associated with amyloidosis.

**Sequence Variants at Protein Level Including 18 a.a. Signal Peptide (Three-Letter Code, Sequence) ***	**References**	**Sequence Variants at Protein Level (Single-Letter Code)**	**Sequence Variant at ** **cDNA Level**	**Exon**
p.Tyr72Asn	[10]	Y54N	c.214T > A	2
p.Ile74Thr	[4]	I56T	c.221T > C	2
p.Phe75Ile	[11]	F57I	c.223T > A	2
p.Trp82Arg	[3]	W64R	c.244T > A	2
p.Asp85His	[4]	D67H	c.253G > C	2
p.Asp85Gly	[12]	D67G	c.254A > G	2
p.Thr88Asn **	[6]	T70N	c.263C > C	2
p.Thr88Asn/Trp130Arg	[7]	T70N/W112R	c.263C > A, c.388T > C	2, 4
p.Leu102Ser	[5]	L84S	c.305T > C	3
p.Asp105Gly	[9]	D87G	c.314A>G	3
p.Trp130Arg **	[7]	W112R	c.388T > C	4

* According to the Human Genome Variation Society (HGVS). ** The substitutions p.Thr88Asn and p.Trp130Arg are not amyloidogenic but belong to the polymorphism variant.

This type of amyloidosis has a very broad spectrum of clinical manifestations, depending on the location and volume of pathological deposits of lysozyme, which may affect all organs and tissues, with the exception of the central nervous system. Involvement of the arterial vessels, GIT, liver, kidneys, spleen, adrenal glands, thyroid, lacrimal and salivary glands, lymph nodes, peripheral nervous system, skin, and bone marrow are typical.

One of the most striking and typical clinical manifestations is the defeat of the GIT. Involvement of the upper GIT often leads to impaired motor function, which is manifested by reflux esophagitis and vomiting with bile [13,14]. However, even more significant for the diagnosis is the tendency to bleed in different parts of the GIT caused by ulcerative lesions of the mucous membrane and vascular lesions. Cases of erythematous gastritis, peptic hemorrhagic ulcers, and rectal bleeding have been described; notably, an accumulation of amyloid deposits in the colonic mucosa may mimic ulcerative colitis. Hemorrhagic bleeding makes this type of amyloidosis a very malignant disease. The literature describes cases of death from recurrent hemorrhages in the mesentery of the small intestine, including those caused by rupture of altered mesenteric lymph nodes, followed by intra-abdominal bleeding. There are also known cases of severe liver damage with the formation of subcapsular hematomas [15], followed by its rupture and development of hemoperitoneum [16].

Another of the clinical manifestations is skin lesions in the form of various manifestations of hemorrhagic syndrome, from petechiae to bruising.

The literature describes cases of bleeding of mucous membranes outside the GIT, including bronchial mucosa. In 2017, a case of lysozyme amyloidosis affecting the lungs and bronchi was described. The patient, along with the lesion of the GIT, developed hemoptysis caused by the bronchial mucosa bleeding. The accumulation of amyloid in the tissues was confirmed by endobronchial biopsy [13].

Kidney damage is characterized by slow development, with the gradual formation of nephrotic syndrome, nephrogenic arterial hypertension, and chronic renal failure (CRF), often leading to the need for hemodialysis at a relatively early age. Histological analysis reveals amyloid deposits both in the glomeruli and in the interstitium; the amount of deposits is variable.

Due to damage to the conjunctiva and accessory lacrimal glands, dry eye syndrome is one of the earliest manifestations of this type of amyloidosis. It may be accompanied by a decrease in visual acuity. Dry mouth syndrome can also occur if the salivary glands are affected. It should be noted that lysozyme is an essential component of the secretion of the lacrimal and salivary glands. Apparently, the active production of lysozyme explains the frequent involvement of these organs.

Involvement of the heart is considered rare, but over the past five years, cases of lysozyme amyloidosis accompanied by damage to the cardiovascular system with hypertrophic cardiomyopathy and pericardial effusion have been described [10].

Other manifestations of lysozyme amyloidosis include infiltration with amyloid deposits of the spleen (with subsequent splenomegaly), nerve ganglia, adrenal glands [16,17], thyroid gland [4,13], arterial vessels (including the temporal artery), and bone marrow [10,13].

## 2. Results

### 2.1. Patients and Clinical Observations

Woman, 65 years old (patient 1). She has been observed since 1997, when, at the age of 46, she was diagnosed with spontaneous GI bleeding and the formation of spontaneous hematomas in the abdominal cavity. The platelet count was 177,000 per µL, and no abnormalities in the hemostasis system were detected. Ultrasound revealed an enlarged spleen and retroperitoneal lymph nodes. There were no significant deviations in the bone marrow trephine. A biochemical blood test revealed signs of hepatocytolysis (AST 99 U/L, ALT 117U/L) and cholestasis (ALP 1078 U/L, GGT 455 U/L). Proteinuria in the range of 0.8–1 g/day was also detected; an immunochemical study of urine revealed the presence of albumin, which indicated damage to the glomerular filter. Due to the recurrence of bleeding, mainly GI, the patient was repeatedly examined in a hematological hospital; pathology of hemostasis, lymphoproliferative diseases, myeloma, and other hemoblastoses were excluded. In 2011, a skin biopsy at the site of the hematoma was performed, which revealed amyloid. The patient had no signs of heart damage; nephropathy was not typical–despite the potential risk of developing nephrotic syndrome, there was no tendency to further develop proteinuria. There were no signs of activation of acute-phase markers of inflammation. Monoclonal gammopathy was ruled out by immunochemical studies with the use of highly sensitive immunofixation methods and quantification of free light chains of immunoglobulins. Immunophenotyping of bone marrow cells was also carried out; no aberrant plasma cells were found. According to the results of a molecular genetic study, transthyretin mutations were excluded. Hence, the prerequisites for AA, AL, or ATTR amyloidosis were reliably excluded. Similar clinical manifestations were noted, however, in mother and daughter (patient 2). Spontaneous skin hemorrhages were also noted in the 17-year-old grandson of patient 1. Thus, the observed disease was characterized by an autosomal dominant type of inheritance and was accompanied by hemorrhagic complications affecting the GIT. Taking into account the family history of the disease and the exclusion of other most common variants of amyloidosis, hereditary lysozyme amyloidosis was diagnosed. In 2016, the patient died as a result of multiple organ failure that developed after massive gastric bleeding. The patient’s death occurred from generalized amyloidosis with damage to all organs and systems except the central nervous system, complicated by the development of atrophy of internal organs, recurrent bleeding from the nasal mucosa and gastrointestinal tract, anemia, disseminated intravascular coagulation syndrome, and symptoms of multiple organ damage.

The daughter (patient 2) of patient 1, aged 41 in 2021, notes hematomas spontaneously occurring for no apparent reason, dry mouth, dry skin, and swollen lymph nodes of all groups. She was diagnosed with amyloidosis in 2017 (at the age of 37) after a histological examination of biopsy specimens of subcutaneous adipose tissue and colon mucosa.

### 2.2. Histological Examination of Biopsy and Autopsy Material

Tissue samples of both patients were subjected to histological examination (with an additional study to detect amyloid). The following tissue samples (materials) were studied: gastrobiopsy specimens of the stomach of patient 1, biopsy of subcutaneous adipose tissue (SAT) of the abdomen and rectal biopsy of patient 2, autopsy material of patient 1 (brain tissue, heart, aorta, lungs, kidneys, bladder, spleen, lymph node, pancreas, liver, various parts of the GIT, uterus, ovary, thyroid gland, adrenal gland, skin, soft tissues of the thigh and pancreas). In addition to verifying amyloidosis and determining the spread and the severity of organ involvement, an important task was to establish the type of amyloid. To determine the type of amyloid, histochemical (test with potassium permanganate) and immunohistochemical studies were initially carried out, but the results were inconclusive.

#### 2.2.1. Patient 1

A stomach biopsy was performed and confirmed the presence of amyloid deposits in the mucosa.

According to the results of the analysis of the autopsy material, damage to all organs and systems, with the exception of the central nervous system, was revealed. A massive lesion of the GIT, cardiovascular system, spleen, and kidneys was observed.

The abundance of deposits led to the subtotal replacement of the spleen tissue, while macroscopically, the spleen was enlarged in size and weight (894 g), had the appearance of a “lardaceous spleen”, and was dense and fragile to the touch. Microscopic examination revealed preserved elements of white pulp (lymphoid tissue) in only a few fields of view (Figure 2A).

The adrenal glands were macroscopically enlarged, brownish-gray, of a dense consistency, while the tissue was fragile and crumbling; microscopic examination shows subtotal replacement of the parenchyma with amyloid deposits and pronounced atrophic changes in intact areas (Figure 2B).

Microscopic examination of the kidneys revealed pronounced atrophic tissue changes, with an abundance of amyloid deposits in the interstitium (much more in the cortical layer than in the medulla). Only a small part of the glomeruli was affected by amyloids (total replacement of single glomeruli, segmental replacement of a small part). (Figure 2C,D). Macroscopic examination showed that the kidneys were enlarged (weight of the left kidney 282 g, weight of the right kidney 246 g), very firm to the touch, waxy due to a change in texture, and had a distinct yellowish tinge of the cortical layer.

Massive amyloid deposits were observed in the samples from various parts of the GI tract of all layers of the wall. The abundance of deposits in the lamina propria led to atrophic changes and mucosal ulceration. (Figure 2E). Remarkable is the partial or complete replacement of smooth muscle tissue in the muscularis mucosae and partial replacement in the muscularis propria.

Massive deposits in the interstitium and atrophic changes were observed in the thyroid samples.

In the pancreas, accumulation of amyloid deposits in the blood vessels and interstitium was revealed, and pronounced atrophic changes of the acinar and insular parts were shown. At the same time, just a small part of the islets of Langerhans was totally replaced by amyloid mass.

The liver was reduced in size and weight (weight 1132 g), dense to the touch, brownish-red, with a yellowish tint, and a greasy luster on the incision. Microscopic examination revealed deposits of amyloid in the capsule, perisinusoidal space (lesion focal, weak), and stroma of the portal tracts (an abundance of deposits was observed in combination with atrophic changes in the triads).

Examination of the heart revealed cardiomegaly (heart weight 462 g), a brownish-red color of the myocardium with an abundance of small (up to 0.1–0.2 cm in diameter) diffusely located grayish inclusions, thickening of the left ventricular wall up to 1.8 cm, the presence of stable atherosclerotic plaques with no significant stenosis of the coronary arteries. Micropreparations of heart tissue revealed a diffuse-nodular form of myocardial damage and the presence of amyloid deposits in the endocardium and epicardium. (Figure 2F). It should be noted that in each of the studied organs, including the heart, aorta, lungs, kidneys, bladder, spleen, lymph node, pancreas, liver, esophagus, stomach, small intestine, large intestine, uterus, ovary, thyroid gland, adrenal glands, skin and soft tissues of the thigh, deposits of amyloid were found in the walls of blood vessels. Vascular obstruction by amyloid deposits was not observed.

Moreover, amyloid deposits were found in the endonervium and in the perinervium of the nerve trunks of the skin, soft tissues, and internal organs.

#### 2.2.2. Patient 2

A histological study of subcutaneous adipose tissue (SAT) of the abdomen and colon biopsies was carried out.

Numerous amyloid deposits along the adipocyte membranes, grade CR3+, CR4+ (based on the visual scale of [18]), and a small number of deposits in the walls of microvasculature vessels were detected in SAT; both squashed native preparations and sections from paraffin blocks were examined.

The study of the colon endoscopic biopsies revealed multiple amyloid deposits in the lamina propria, muscularis mucosae, and the walls of the microvasculature vessels (Figure 2G,H).

### 2.3. Mass Spectrometry Analysis

To determine the amyloid-forming protein and, consequently, the final determination of the type of amyloidosis, a mass spectrometry analysis of biopsy (SAT) and autopsy (SAT, kidney, spleen) material with histologically confirmed deposits was carried out.

#### 2.3.1. Patient 1

In the preparation of frozen subcutaneous adipose tissue after total protein extraction, along with the main structural tissue proteins, components typical of the structure of amyloid deposits were found: serum amyloid p-component (SAP), apolipoproteins AI (Apo-AI), AIV (Apo-AIV), E (Apo-E), and heparan sulfate proteoglycan (Table 2). According to [19], the presence of SAP, Apo-AIV, and Apo-E with a sensitivity of 88% and a specificity of 96% (odds ratio 36.3) allows us to confirm by mass spectrometry that the analyzed proteinograms were taken from the amyloid deposits. This is also consistent with the histological findings.

Along with the aforementioned amyloid marker proteins, the lysozyme was also determined with a high degree of reliability (determined by the index −10lgP ≥ 20), which indicated its fundamental importance in the formation of amyloid deposits, since other significant proteins, except for Apo-AI, are nonspecific components of any type of amyloid.

The p.T88N amino acid substitution in the lysozyme sequence was also revealed by mass spectrometry (identification was carried out in the PEAKS Studio program), taking into account post-translational modifications or amino acid substitutions (PEAKS PTM algorithm) (Figure 3).

The same components of amyloid were detected along with structural proteins in kidney tissue from a glass preparation and in the FFPE tissue samples of the kidney and spleen.

#### 2.3.2. Patient 2

In the subcutaneous adipose tissue biopsy, mass spectrometry revealed the same proteins as in the tissue samples of patient 1: lysozyme, SAP, Apo-E, Apo-AIV, heparan sulfate proteoglycan, and Apo-AI (Table 3).

Table 3 shows proteins characteristic of amyloid deposits extracted from the histological preparation of patient 2 (daughter). Along with a specific protein that determines the type of amyloidosis (lysozyme with the amino acid substitutions p.F21L/T88N) (Figure 4), amyloid signature proteins were identified in the samples with a high degree of reliability (−10lgP ≥ 20). In particular, a number of serum proteins are present in the amyloid deposits.

We also studied the proteomes of kidney and subcutaneous tissue (autopsy samples) obtained from “healthy donors”, with histologically confirmed absence of amyloid deposits. No specific amyloidogenic proteins or marker amyloidosis proteins have been identified in these preparations.

In the samples of patient 2, as well as in patient 1 samples, lysozyme was determined with high reliability and almost complete coverage of the amino acid sequence, which made it possible to identify various amino acid substitutions. Thus, in the samples of patient 2, along with the p.T88N amino acid substitution (Figure 3 and Figure 4), which was detected in the tissue samples of patient 1, a peptide with the p.F21L substitution was also detected by mass spectrometry. This substitution involves the initial fragment of the amino acid sequence after the cleavage of the signal peptide. This part of lysozyme is the most technologically unstable. In particular, in the samples of patient 1, the p.F21L amino acid substitution was not detected. Attempts to determine the amino acids typical of positions 29 and 30–threonine and leucine–in all samples were also unsuccessful. Therefore, despite the high level of coverage of the amino acid sequence, we performed sequencing of the lysozyme encoding gene to confirm the p.F21L mutation. Since patient 2 had two children at the time of the study, it was also necessary to establish their inheritance of this type of lysozyme (Figure 4). Therefore, the analysis of the nucleotide sequence of the LYZ gene was also carried out for them.

### 2.4. Genetic Testing

We have carried out sequencing of the *LYZ* gene of the proband (patient 2) and her children. Nucleotide substitutions in the first (AAA → GAA) and second (TGG → TCG) exons, which led to p.F21L/T88N replacement in the amino acid sequence of lysozyme, were identified in the genes of the proband and her son (Figure 5). However, no mutations were revealed in the daughter *LYZ* gene. The proband and her son are heterozygous for this trait (Figure 5). The p.F21L substitution (exon 1) was not found in the ESP (https://evs.gs.washington.edu/EVS/, accessed on 8 July 2022) and gnomAD databases and is not related to polymorphism.

Using the results of genetic and proteomic analysis and the obtained clinical data, we compiled a pedigree chart (tree) of amyloidosis inheritance in this family (Figure 6).

### 2.5. Structural Analysis

Chicken lysozyme was the first enzyme structure to be obtained using X-ray crystallography by D.C. Phillips in 1965. It is interesting to look at the structural properties of this protein in terms of various properties, ranging from amyloidogenic regions, changes in protein stability, and ending with antimicrobial regions since the protein has antibactericidal properties.

Lysozyme is a protein with antibacterial activity. It is widely distributed in various tissues and body fluids. It belongs to the class of glycosidic hydrolases. The activity of lysozyme is to act on the peptidoglycan of the bacterial cell wall, which consists of glycan chains of various N-acetylglycosamines and N-acetylmuramic acid, which are cross-linked with peptides through the N-acetylmuramic acid sites. This enzyme is called 1,4-β-d-N-acetyl muramidase, which catalyzes the cleavage of the glycosidic bond between the first carbon of N-acetylmuramic acid and the fourth carbon of N-acetylglucosamine, resulting in cell wall breakdown. Lysozyme converts the sugar muramic acid into a tense conformation, and with the combined action of two key elements–glutamic acid at position 35 and aspartic acid at position 53–hydrolyzes glycosidic bonds.

If we look at the amino acid composition of this protein, we should note the high content of some residues in comparison with the human proteome composition of amino acids. Lysozyme contains twice as many amino acids as asparagine and arginine and almost three times more cysteine and tryptophan (Figure 7). However, there is two times less phenylalanine and serine, three times less glutamic acid and histidine, and four times less proline.

The low number of prolines makes this protein flexible, especially considering that there are slightly more glycines (1.3 times) and alanines (1.5 times) compared to the proteome composition (Figure 7). This is also demonstrated by deuterium-hydrogen exchange data that chicken lysozyme exchanges more protons (134, the average number of protons exchanged, measured by mass spectrometry, first 15 min) than human lactalbumin (84), which is similar in mass [20]. Four disulfide bonds take on the role of prolines, holding the structural framework and preventing the rearrangement of the native structure (Figure 1).

Almost all substitutions except one (D67G) result in a small decrease in accessible surface area (Table 4) and a small change in protein stability (Table 4). Another parameter, scaled solubility, and pI, also changes slightly.

Phenylalanine number 3 is not included in the aromatic network of contacts, which is made by four pairs [21]: Trp28-Phe57; Trp34-Tyr124; Phe57-Trp109; Tyr63-Trp64. Therefore, the replacement of aromatic residues Phe57 and Trp64 will lead to the destabilization of the protein (Table 4).

Using the Antimicrob program, four antibacterial sites are predicted in the lysozyme sequence (10–15; 76–83; 94–99; 107–111) [22].

**Table 4 ijms-24-14453-t004:** Calculated accessible surface area (ASA) residue before and after mutation, scaled solubility, pI, and change in thermodynamic stability upon mutation.

Mutation	ASA before Mutation % from Maximum ASA [23]	ASA after Mutation % from Maximum ASA [23]	Predicted Scaled Solubility [24]	pI [24]	ΔΔG kcal/mol [25]
Y54N	9	8	0.571	10.71	0.208
I56T	1	1	0.565	10.58	0.212
F57I	1	4	0.581	10.58	0.076
W64R	21	10	0.579	10.73	0.185
D67H	17	15	0.539	10.74	0.273
D67G	17	31	0.568	10.74	0.147
T70N	4	4	0.575	10.58	−0.046
T70N/W112R	4/10	4/2	0.591	10.73	*/*
L84S	2	2	0.561	10.58	−0.122
D87G	42	33	0.568	10.74	0.186
W112R	10	2	0.579	10.73	0.965
F3L/T70N	3/4	2/4	0.599	10.58	0.154/−0.046

* The ABYSSAL program cannot calculate double mutations.

Prediction of amyloidogenic sites by five different programs shows that almost all mutations fall within the range of these predictions (Table 5). In the paper [9], it has been demonstrated that all previous mutations do not greatly affect the prediction of amyloidogenic properties. Our mutation also does not change the prediction of amyloidogenic sites.

Our analysis shows that, in general, none of the mutations leads to catastrophic changes in the parameters we calculated. This was also emphasized in a previous article where the last mutation in lysozyme (D87G) was discovered [9].

## 3. Discussion

Convincing evidence of an amyloid-forming protein is considered to be its absolute quantitative predominance in the amyloid composition. This requires electrophoretic separation of the amyloid protein fraction, which is not always amenable to convincing standardization [30,31,32]. It is particularly remarkable that in this work, the dominance of lysozyme was determined by indirect evidence: the high reliability of its identification and the completeness of the amino acid sequence recovered from fragments in the amyloid extract. Another amyloid-forming protein that reached the required level of reliability index was only apolipoprotein A1. However, its amino acid sequence was not restored sufficiently, and the protein itself was not reproduced in all samples. The importance of lysozyme in the process of amyloidogenesis is confirmed by the results of a genetic study in the family, which showed a stable aberration of its amino acid sequence for p.F21L/T88N mutations.

This method of amyloid typing cannot be recommended as a standard for routine practice. However, its reliability in this case is beyond doubt. In this context, some general problems in disease diagnosis become apparent. Pathognomonic symptoms sufficient for a reasonable etiologically verified diagnosis are extremely rare in clinical practice. Much more often, several different methods, often duplicating each other, are used to make a final diagnosis. For example, patient 1 had to undergo a highly sensitive immunoassay coupled with bone marrow immunophenotyping and, subsequently, immunohistochemistry to rule out plasma cell dyscrasia (required for the development of AL amyloidosis). Immunohistochemistry has certain limitations; in particular, the results of using antisera to various amyloid protein precursors may be erroneous due to changes in the antigenic properties of the amyloid precursors. The analysis of clinical data in combination with the results of proteomic analysis of material with histologically confirmed deposits of amyloid is optimal. However, mass spectrometry may also have limitations. As an example, the identification of apolipoprotein A1 in the amyloid composition of both patients is a potential prerequisite for the diagnosis of another type of amyloidosis. Moreover, if the content of lysozyme and apolipoprotein A1 in the amyloid composition is comparable, the quantitative criterion in diagnosing the type of amyloidosis will lose its reliability. Observation of patients with hereditary ATTR-amyloidosis shows that several years after liver transplantation, normal non-mutant transthyretin begins to dominate over mutant transthyretin in amyloid composition. The same regularity was revealed in experimental models with transplantation of casein AA amyloidosis in animals. Thus, the initial amyloid deposits are capable of realizing the amyloidogenic potential of even normal precursor proteins, acting as an amyloid seed (or amyloid accelerating substance) in the manner of a crystalline phase in saturated solutions. Apparently, the involvement of apolipoprotein AI in the process of lysozyme amyloidogenesis and, accordingly, its identification in the amyloid composition in tissue samples of the examined family occurred in this way. Supplementing the results of mass spectrometry with data from a genetic study of the family made it possible to avoid potential systematic errors and, therefore, can be recommended as a useful option for proteomic analysis of all patients with suspected hereditary forms of amyloidosis.

The results of clinical analysis and morphological data together revealed a typical and well-known pattern for systemic amyloidosis–a significant clinical and morphological dissociation with extensive histological lesions of all organs except for the central nervous system (according to the autopsy data of patient 1). Clinical manifestations were mainly limited to hemorrhagic syndrome and minor signs of the spleen, liver, and kidney involvement. Despite the damage to the glomeruli, the nephrotic syndrome, characteristic of other amyloidosis forms, did not develop in the patient.

Mass spectrometry also gives rise to discussions about clinical and morphological dissociations in another aspect. In the alliance of marker proteins (characteristic of amyloids of any type) identified by mass spectrometry, SAP deposition is noteworthy. These proteins belong to the pentraxin family and, through a calcium-dependent mechanism, can bind to a wide range of ligands, similar to lectins [33]. This receptor polyvalence of SAP apparently makes it possible to isolate the functionally active groups of proteins in the amyloid composition. In addition, the special affinity of SAP for lipoproteins provides the sorption of different types of apolipoproteins in amyloid: Apo-AIV and Apo-E, forming an amyloid marker ensemble, as well as Apo-AI and, possibly, SAA, potentially limiting the diagnostic capabilities of mass spectrometry per se. Of particular importance is the biological utility of fixing amyloid with SAP and other marker proteins: in this way, the body avoids the destructive effect of amyloid on the function and structure of the affected tissue [32,34]. This may explain the clinical and morphological dissociation between the number of actually affected tissues and the formed dysfunctions of these tissues, which is characteristic of amyloidosis. Such a reaction to the relatively safe (up to certain limits, after which amyloidosis turns into a pathological process) isolation of large volumes of cell-free masses apparently has ancient evolutionary prerequisites and a stable functioning machinery. There are numerous examples of plants, invertebrates, and primitive chordates with amyloid-like accumulations that perform various functions–food, protective, or even predatory–mixins (hagfish) cover the respiratory organs of prey fish with amyloid [35]. Even unicellular organisms can synthesize amyloid: enterobacteria and yeasts are able to isolate themselves from the aggressive environment with amyloid biofilms, thus exhibiting significant antibiotic resistance. Functionally normal amyloid can be detected even inside the cell. This achieves reliable localization and regulation of various biochemical reactions–the cryopyrin inflammasome provides the production of IL-1, and the Pmel protein is the basis for the synthesis and, at the same time, protection from the toxic effects of melanin in melanocytes [35]. In practice, it seems difficult and unpromising to suppress these mechanisms.

It should be mentioned that the absence of amyloid deposits in the tissue does not necessarily indicate that the protein is non-amyloidogenic [36]. But in this work, we focus on those proteins that form amyloid deposits that are associated with the development of systemic or local amyloidosis. To date, 42 such proteins have been classified, determining the manifestation of this pathology [37].

Clinical experience shows that when remission is achieved in patients with amyloidosis, even in organs that have restored their (normal) functioning, there are still large amounts of amyloid deposits in their tissues. Are attempts to treat patients with amyloidosis by dissolving amyloid deposits justified in this case? Such attempts are more likely to disrupt the fragile mechanisms of regeneration in the affected tissue, releasing the biological aggressiveness of the precursor proteins that make up the amyloid. It is possible that the inclusion of proteomic analysis data in clinical and morphological comparisons will make it possible to answer a wide range of questions in the future.

## 4. Materials and Methods

### 4.1. Histology Investigations

Both autopsy and biopsy materials were fixed in a 10% neutral buffered formalin solution and embedded in paraffin according to standard protocols. Native preparations of abdominal subcutaneous adipose tissue were made by squashing small fragments between glass slides. In addition, the SAT fragments of both patients were frozen (storage was carried out at −18 °C) without preliminary fixation with formalin.

Paraffin sections, in addition to the standard staining with hematoxylin–eosin, were stained with Congo red and examined in polarized light. Native preparations were also Congo red stained and examined in polarized light. Amyloid deposition was determined by birefringence with a characteristic luminescence spectrum (greenish and yellowish-greenish hues) of congophilic structures. The studies in polarized light were performed on a specialized microscope LeicaDM2000 (Wetzlar, Germany).

### 4.2. Mass Spectrometry

#### 4.2.1. Extraction of Total Protein

Extraction of total protein from the SAT samples of patients 1 and 2 was performed according to the following procedure. Frozen subcutaneous fat samples were homogenized in lysis buffer (4% SDS, 100 mM β-mercaptoethanol) using Miulab MT-30K manual homogenizer for 5 min at maximum power. Then, the sample was incubated at +4 °C for 24 h. After a day of sample incubation, the total protein was precipitated with acetone. Precipitates were redissolved in the buffer for loading into a polyacrylamide gel. Electrophoretic separation of proteins was carried out in a 15% polyacrylamide gel in a Tris/Glycine/SDS gel buffer system (a Mini Gel Tank (Thermo Fisher Scientific, Waltham, MA, USA) and a vertical electrophoresis chamber was used).

Extraction of total protein from FFPE tissue samples (spleen and kidney samples of patient 1) was carried out according to the same protocol, except for incubation of the tissue homogenate at 85 °C for 30 min, which periodically interrupted the homogenization process. To obtain the total protein, the method of precipitation of proteins with ethanol was used. The total protein was obtained by the precipitation with ethanol. Then, polyacrylamide gel electrophoresis was also carried out.

#### 4.2.2. Independent Hydrolysis by Three Proteases with Different Specificities in a Gel

The procedure was carried out according to the standard protocol [38]. Proteolysis was carried out using three proteases–trypsin, chymotrypsin, and proteinase K.

#### 4.2.3. HPLC-MS/MS–Analysis

Total protein hydrolysates were analyzed by reverse-phase high-performance liquid chromatography and mass spectrometry (HPLC-MS). An EASY-nLC 1000 chromatograph and an OrbiTrap Elite mass spectrometer (Thermo Scientific, San Jose, CA, USA) were used. Chromatographic separation of peptides was carried out on a capillary column manufactured under laboratory conditions: column length 45 cm, diameter 75 µm, Aeris PEPTIDE XB-C18 3.6 µm with a pore size of 100 Å was used as a phase. Separation of the peptides was carried out in a gradient of acetonitrile from 0 to 50% for 190 min at a flow rate of 250 nl/min.

The obtained mass spectrometry data were analyzed using the PEAKS Studio-7.5 program (Bioinformatics Solutions Inc. (BSI), Waterloo, ON N2L 6J2, Canada).

### 4.3. Sequencing of the LYZ Gene

Genomic DNA was isolated from buccal epithelium samples from patient 2 and her son and daughter using the «ExtractDNA Blood» kit (Evrogen, Moscow, Russia). Amplification of DNA fragments containing exons of the *LYZ* gene was performed using the corresponding pairs of oligonucleotides (Table 6) and KOD Hot Start DNA Polymerase (Merck, Darmstadt, Germany) according to the manufacturer’s recommendations. Determination of the nucleotide sequence of PCR products was carried out by the Sanger method.

## 5. Conclusions

We have described for the first time a case of autosomal dominant hereditary amyloidosis in a Russian family caused by the deposition in organs and tissues of lysozyme with a double amino acid substitution p.F21L/T88N. A preliminary diagnosis of lysozyme amyloidosis was formulated after a clinical analysis of the manifestations: hemorrhagic syndrome as a dominant clinical sign, an autosomal dominant inheritance of the disease, and the absence of the necessary prerequisites for the development of other more frequent forms of systemic amyloidosis. However, this type of amyloidosis was finally diagnosed after mass spectrometric analysis. In conclusion, we emphasize that the p.F21L lysozyme mutation identified in this family and its amyloidogenicity have been described for the first time and, therefore, are of particular scientific value.

## Figures and Tables

**Figure 1 ijms-24-14453-f001:**
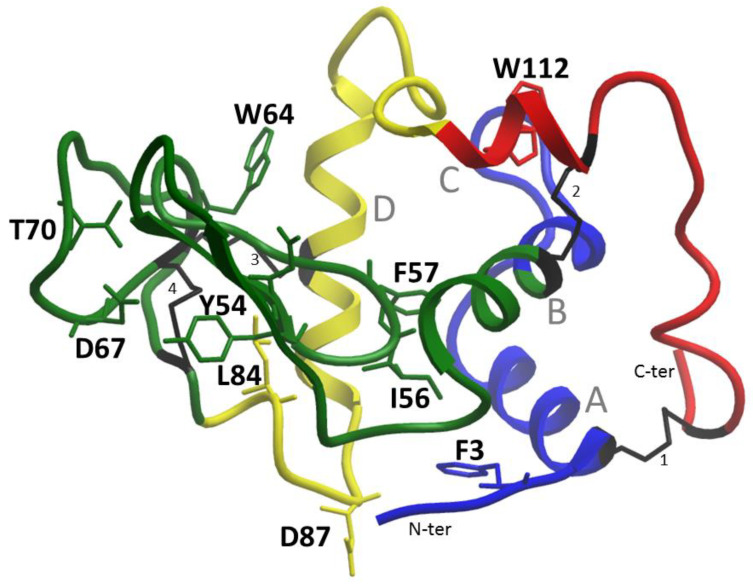
Amino acid substitutions in the three-dimensional structure of lysozyme (code pdb 1REX), leading to the development of ALys amyloidosis. The region from 1 to 28 a.a. is shown in blue (1–46 encoded by exon 1, since the structure is without a sequence corresponding to the signal peptide), encoded by exon 1, the region from 29–82 (47 to 100 a.a., exon 2) marked in green, the area from 83–110 (101 to 128 a.a., exon 3) marked in yellow, region from 111–130 (129 to 148 a.a., exon 4) marked in red. The newly identified mutation in this work, F3L, is also shown in blue according to the exon color. The structure of lysozyme consists of β-structural and α-helical domains (α-helices are marked in the figure with letters A–D). There are four disulfide bonds in a protein molecule. Cysteines forming these disulfide bonds are marked in grey. The structure was drawn by the Molsoft ICM 3.5 program.

**Figure 2 ijms-24-14453-f002:**
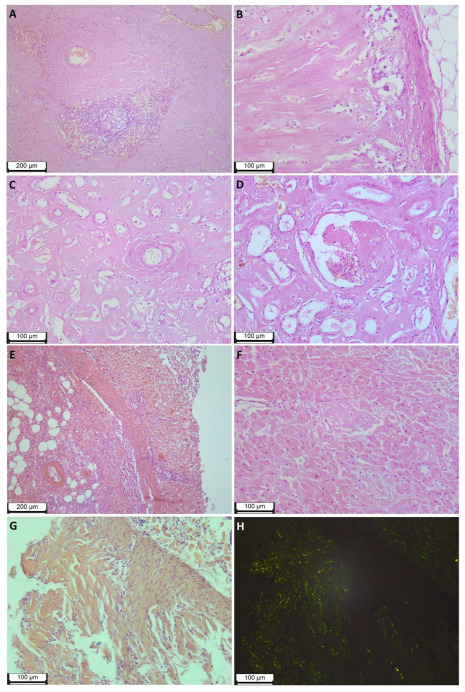
Histological findings of the patient 1 tissues. (**A**)—subtotal replacement of spleen tissue with amyloid deposits, stained with hematoxylin–eosin, magnification 100×. (**B**)—subtotal replacement of adrenal tissue with amyloid deposits, hematoxylin–eosin staining, magnification 200×. (**C**)—the medulla of the kidney with an abundance of deposits in the interstitium, hematoxylin–eosin staining, magnification 200×. (**D**)—cortical layer of the kidney, partial replacement of the glomerulus with deposits, accumulation of deposits in the interstitium, hematoxylin–eosin staining, magnification 200×. (**E**)—stomach, massive deposits of amyloid in the mucosa and submucosa, Congo red staining, magnification 100×. (**F**)—heart, amyloid deposits in the interstitium, hematoxylin–eosin staining, magnification 200×. (**G**)—numerous deposits of amyloid in the muscularis mucosa of the colon of patient 2, stained with Congo red, magnification 200×, bright-field microscopy, (**H**)–polarized light examination for sample from (**G**).

**Figure 3 ijms-24-14453-f003:**
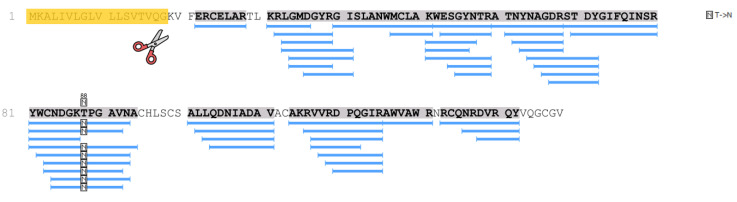
Mass spectrometry determination of the amino acid substitution in patient 1 lysozyme sequence (SAT autopsy sample). Coverage of the amino acid sequence of lysozyme (PeaksStudio program, Spider mode). Recognized peptides are highlighted in blue, and the signal peptide sequence (amino acid residues 1–18), which is cleaved off during lysozyme processing, is highlighted in orange.

**Figure 4 ijms-24-14453-f004:**
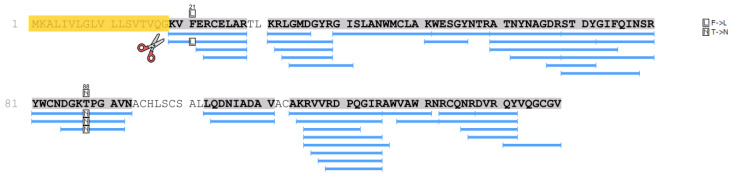
Mass spectrometry determination of the amino acid substitution in patient 2 lysozyme sequence (SAT biopsy sample). Coverage of the amino acid sequence of lysozyme (PeaksStudio program, Spider mode). Recognized peptides are highlighted in blue, and the signal peptide sequence (amino acid residues 1–18), which is cleaved off during lysozyme processing, is highlighted in orange.

**Figure 5 ijms-24-14453-f005:**
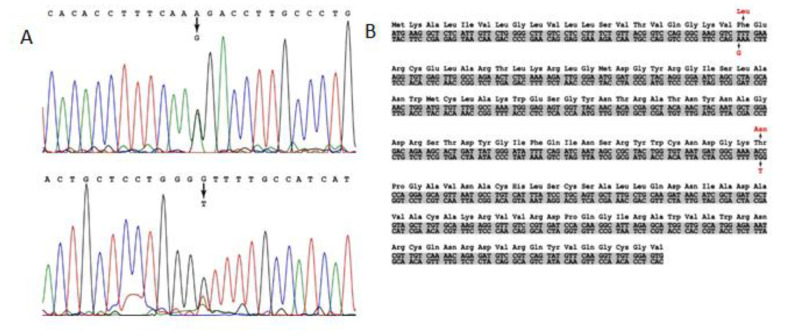
Detection of a mutation in the *LYZ* gene nucleotide sequence for patient 2. (**A**) shows 61 A > G nucleotide substitution resulting in p.F21L amino acid substitution and shows 263 G > T nucleotide substitution resulting in p.T88N substitution in the lysozyme amino acid sequence. (**B**) shows the nucleotide sequence in the gene, the nucleotide substitutions that led to the amino acid substitutions in the lysozyme sequence are highlighted in red.

**Figure 6 ijms-24-14453-f006:**
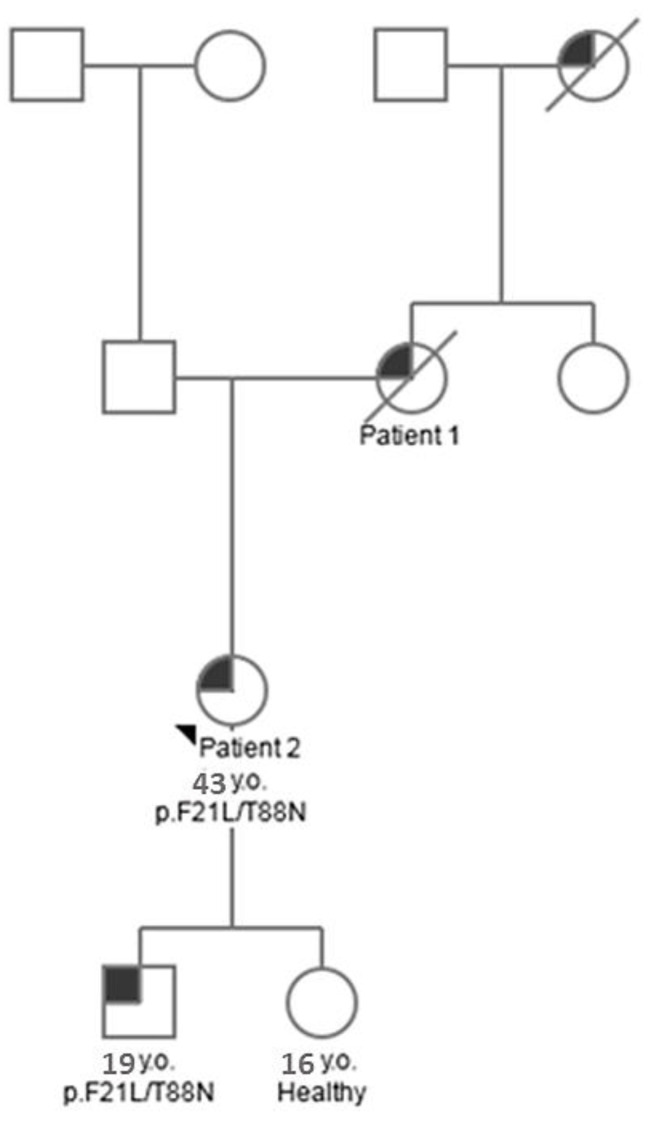
Pedigree describing the inheritance of ALys-type amyloidosis in the described family. The arrow indicates the proband (patient 2) in relation to which the pedigree was compiled. The grey sector marks the sign–ALys amyloidosis. The crossed-out symbol represents the relatives of the proband who died at the time of the pedigree compilation.

**Figure 7 ijms-24-14453-f007:**
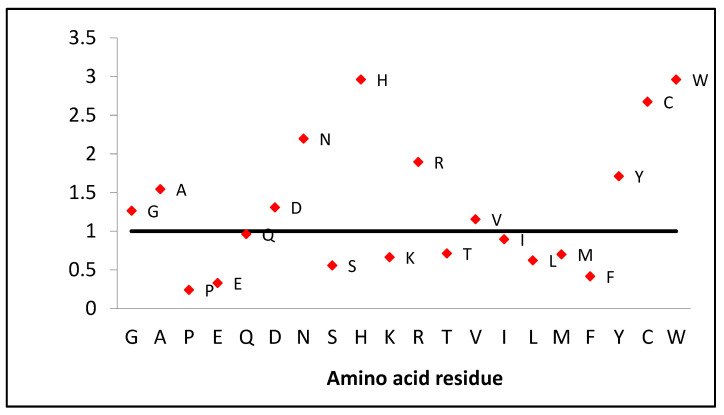
Amino acid composition of human lysozyme. The values of the occurrence of amino acids in the human proteome are taken as one.

**Table 2 ijms-24-14453-t002:** Proteins accumulated in amyloid deposits. Extracted from histological preparations of patient 1 (mother) autopsy material.

Sample	Subcutaneous Adipose Tissue (Autopsy Material)	Kidney (Stained Section, Scrape from Glass Preparation)	Spleen (FFPE Tissue Sample)
Protein	Reliability Parameter (−10lgP) *		
Lysozyme	148	210	211
Serum amyloid p-component	137	180	-
Apolipoprotein E	125	191	87
Apolipoprotein AI	-	97	-
Apolipoprotein AIV	-	195	-
Heparan sulfate proteoglycan	166	151	-

* (−10lgP) is the identification reliability parameter. A protein is considered to be reliably recognized if the value (−10lgP) ≥ 20 is calculated as a weighted sum of the −10lgP estimates of the recognized protein peptides. After removing any redundant peptides, the recognized peptides are sorted by −10lgP score in descending order, and the k-*th* rank peptide contributes to the weighted sum with a weight of 1/k.

**Table 3 ijms-24-14453-t003:** Proteins determined by mass spectrometry analysis of the biopsy of the SAT (subcutaneous adipose tissue) of patient 2.

Protein	−10lgP
Lysozyme	216
Serum amyloid p-component	195
Apolipoprotein E	249
Apolipoprotein A-I	244
Apolipoprotein A-IV	223
Heparan sulfate proteoglycan	401

**Table 5 ijms-24-14453-t005:** Regions of the amino acid sequence of lysozyme predicted as amyloidogenic using five programs.

Programm	Sequence	Start	End	Length
Aggrescan [26]	MKALIVLGLVLLSVTVQGK	1	19	19
	LANWMCLAK	43	59	9
	YGIFQINSR	72	80	9
	ACHLSCSAL	94	102	9
	RAWVA	125	129	5
APPNN (R pocket)	ALIVLGLVLLSVTVQG	3	18	16
	ISLANWMCLA	41	50	10
	ATNYNA	60	65	6
	TDYGIFQINS	70	79	10
	AVNACHLSCSALLQDNI	91	107	17
	QYVQGCGV	141	148	8
FoldAmyloid [27]	KALIVLGLVLLSV	2	14	13
	LANWMCLAK	43	51	9
	GIFQI	73	77	5
	SRYWC	79	83	5
	RAWVAWRN	125	132	8
Waltz [28]	GIFQINS	73	79	7
Pasta 2.0 [29]	MKALIVLGLVLLSVTVQGKVL	1	21	21

**Table 6 ijms-24-14453-t006:** Pairs of oligonucleotides used to amplify DNA containing exons of the *LYZ* gene.

Name	Sequence
ex1_f	GATGTTAAATACTGGGGCCAGCT
ex1_r	CGTATTCCTTTATCGGGTATCTCTG
ex2_f	GCATTACAAAAGGATTCTCTTACAAGTC
ex2_r	GATGTAGTTTTTCTAGGTCCCTCATG
ex3_f	TGCTGAATGGGAGAGAATGAAG
ex3_r	AGCATCTTAAAACTCTAGAAATCCCCT
ex4_f	CTGAGAAATTTGGGTAGGAGTGAAGT
ex4_r	CCATATTTTGCTCCTGCTTCTGT

## Data Availability

The data presented in this study are available on request from the corresponding author.

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
