# Peer review of "First Report of Lysozyme Amyloidosis with p.F21L/T88N Amino Acid Substitutions in a Russian Family"

_ijms, 2023, doi:10.3390/ijms241914453_

Round 1

Reviewer 1 Report

The manuscript titled " First report of lysozyme amyloidosis with p.F21L/T88N amino acid substitutions in the Russian family" submitted by Suvorina MY and coworkers provides a comprehensive clinical description of Lysozime amyloidosis with F21L/T88N substitutions in a Rusian family. Lysozyme amyloidosis is an extremely rare disease characterized by the abnormal aggregation and accumulation of endogenous lysozyme in various tissues and organs of the body. This condition is part of a group of disorders known as amyloidosis, where proteins misfold and aggregate into highly stable amyloid deposits. It is important to note that only thirty cases of this disease have been identified so far. In this study, the authors have provided a detailed description of the familial phenotype, as well as the histological changes observed in the affected individuals with the F21L/T88N mutations. Moreover, the authors used mass spectrometry and DNA sequencing to determine the inheritance of the F21L/T88N mutations. Overall, this is a well-structured and interesting manuscript that significantly advances our understanding of the molecular mechanisms associated to Lysozime amyloidosis in humans. In my oppinion, the manuscript is suitable for publication after major and minor corrections.

Major and minor comments:

1)      Title: Please correct the tittle as follows: “First report of lysozyme amyloidosis with p.F21L/T88N amino 2 acid substitutions in a Russian family”. The use of "the" in the first title suggests that this is the first report of lysozyme amyloidosis specifically within a known or previously mentioned Russian family.

2)      Figure 1. The current structural representation is not clear. It would be interesting if the authors can improve the visibility of the lysozyme mutations. Specifically, they should highlight the two F21L/T88N mutations relevant in the context of present study. The size of the labels should be increased for clarity.

3)      While the clinical and histological manifestations of the disease are well explained, this work lacks biochemical descriptions. For instance, in the case of TTR amyloidosis, the disease associated TTR mutations characterized to date either decrease the tetrameric quaternary structural stability or the monomer’s tertiary structure stability, or both. What is the effect of the mutations on lysozyme? Are mutations affecting the stability of the protein? It would be interesting if the authors can perform stability calculations for these mutations. The author should explain and discuss by which mechanism the mutations promote lysozyme aggregation in these patients. This should be discussed in the manuscript.

4)      Related to my previous comment, the authors should also compare the location of these mutations with the previously described ones. Are the new mutations in buried residues, while the others are solvent exposed? This may explain distinct aggregation mechanisms. This possibility should be discussed.

5)      Line 313: The author mention that thy investigated the “healthy donors”. Please explain whether these patients carried pathologic mutations in the LYS gene and lack the pathological features of the disease, or if the have the WT Lys gene.

6)      Figure 4 and 5: Pleas provide higher quality figures. In the current representation the labels are not readable.

7)      Figure 6 can be included as an additional panel in Figure 5.

8)      Figure 2. Please include size scales in all the images.

9)      Table 1: The current organization of this table is not clear. Please provide references in a separate column.

10)  Line 312-315. The absence of amyloid deposits in the tissue does not necessarily indicate that the protein is non-amyloidogenic. This is often the case of functional amyloids such as hnRNPDL-2 (Nature communications. 2023. 14: 239. 10.1038/s41467-023-35854-0.). This possibility should be considered and discussed.  

English is fine and only minor editing is needed.

Reviewer 2 Report

Authors presented double mutations in lysozyme gene for the amyloidosis and s and showed the interacting proteins and the deposited locations for the systemic manifestation of the disease. 

Few things authors need to show are as following

1. Are double mutations are located in the same pair of the gene or different pairs.

2. Authors need to present the affected rnzymatic functions.

There are few run-on sentences. 

Reviewer 3 Report

This is a case report about lysozyme amyloidosis. It is a rare variant of amyloidosis and difficult to diagnose. The report is much valuable, however there were some issues to be addressed.

# Authors mentioned the patient died due to multiple organ failure derived from GI bleeding. Were there any surgical therapies for GI bleeding such as gastrectomy? In addition, more concise description of clinical course to death should be added.  

# Authors mentioned deposits of amyloid were found in the walls of blood vessels of various organs. Are there any effects of the amyloid deposition of blood vessels in other organs than GI tract such as ischemia?

None

Round 2

Reviewer 1 Report

This is a revised version of the manuscript titled "First Report of Lysozyme Amyloidosis with p.F21L/T88N Amino Acid Substitutions in a Russian Family," submitted by Mariya Yu Suvorina and colleagues. After a thorough review of the revised manuscript, I am pleased to acknowledge that the authors have successfully addressed both minor and major issues that were identified in the initial submission.

Notably, the authors have incorporated additional data and in-depth discussions regarding the biochemistry of human lysozyme aggregation. Furthermore, they have significantly improved figures and they have made changes in the title, introduction, and conclusions sections, which significantly enhanced the quality and clarity of the manuscript.

Considering the comprehensive revisions made in response to my comments and the feedback from other reviewers, I recommend the publication of this manuscript in its current form.

 Only minor editing of English language is required.